# Diabetes Prevalence and Associated Risk Factors among Women in a Rural District of Nepal Using HbA1c as a Diagnostic Tool: A Population-Based Study

**DOI:** 10.3390/ijerph19127011

**Published:** 2022-06-08

**Authors:** Chandra Yogal, Sunila Shakya, Biraj Karmarcharya, Rajendra Koju, Astrid Kamilla Stunes, Mats Peder Mosti, Miriam K. Gustafsson, Bjørn Olav Åsvold, Berit Schei, Unni Syversen

**Affiliations:** 1Department of Clinical and Molecular Medicine, Faculty of Medicine and Health Science, Norwegian University of Science and Technology, 7491 Trondheim, Norway; chandra.m.yogal@ntnu.no (C.Y.); kamilla.stunes@ntnu.no (A.K.S.); mats.p.mosti@ntnu.no (M.P.M.); miriam.gustafsson@ntnu.no (M.K.G.); 2Department of Community Program, Kathmandu University School of Medical Science, Dhulikhel Hospital, Kathmandu University Hospital, Dhulikhel 45200, Nepal; birajmk@gmail.com; 3Department of Gynecology and Obstetrics, Kathmandu University School of Medical Sciences, Dhulikhel Hospital, Kathmandu University Hospital, Dhulikhel 45200, Nepal; sunilashakya@yahoo.com; 4Department of Internal Medicine, Dhulikhel Hospital, Kathmandu University Hospital, Dhulikhel 45200, Nepal; rajendrakoju@gmail.com; 5Regional Education Center, Helse Midt-Norge, 7030 Trondheim, Norway; 6K.G. Jebsen Center for Genetic Epidemiology, Department of Public Health and Nursing, Faculty of Medicine and Health Science, Norwegian University of Science and Technology, 7491 Trondheim, Norway; bjorn.o.asvold@ntnu.no; 7Department of Endocrinology, Clinic of Medicine, St. Olavs University Hospital, 7030 Trondheim, Norway; 8Department of Public Health and Nursing, Faculty of Medicine and Health Science, Norwegian University of Science and Technology, 7491 Trondheim, Norway; berit.schei@ntnu.no

**Keywords:** diabetes, HbA1c, risk factors, instant noodles, women, rural, Nepal, prevalence

## Abstract

Given the scarcity of data on diabetes prevalence and associated risk factors among women in rural Nepal, we aimed to examine this, using glycated hemoglobin (HbA1c) as a diagnostic tool. A cross-sectional survey addressing reproductive health and non-communicable diseases was conducted in 2012–2013 among non-pregnant, married women in Bolde, a rural district of Nepal. HbA1c ≥ 6.5% (48 mmol/mol) was used as diagnostic criterion for diabetes, a cut-off of 7.0% (53 mmol/mol) was used to increase the specificity. HbA1c was measured in 757 women (17–86 years). The prevalence of diabetes and prediabetes was 13.5% and 38.5%, respectively. When using 7.0% as a cut-off, the prevalence of diabetes was 5.8%. Aging, intake of instant noodles and milk and vegetarian food (ns) were associated with increased risk for diabetes. Waist circumference was higher among women with diabetes, although not significant. The women were uneducated (87.6%), and only 12% had heard about diabetes. In conclusion, we observed a higher prevalence of diabetes and prediabetes than anticipated among rural, Nepalese women. The increased risk was mainly attributed to dietary factors. In contrast to most previous studies in Nepal, we used HbA1c as diagnostic criterion.

## 1. Introduction

Low-income countries like Nepal are facing the most rapid shift from communicable to non-communicable diseases (NCDs) and are currently suffering from a double burden of diseases. Type 2 diabetes (T2D) is one of the fastest growing NCDs, comprising around 90% of all diabetes cases worldwide [1]. Major factors driving the rise in T2D are urban residence, aging of the population, poor lifestyle and increasing body mass index (BMI) [2,3]. Poverty, malnutrition, and illiteracy are contributing factors to increased vulnerability among females in low-income countries [4,5]. The prevalence of T2D is increasing particularly in Asia [6,7]. Data on the prevalence of T2D and its risk factors in Nepal are limited, especially in rural districts. Aryal et al. reported an overall diabetes prevalence of 6.9% in a population living at high altitudes in Nepal [8]. In a study from 2018, addressing T2D in semi-urban Nepal, 11.7% had diabetes [9]. This study also reported a prevalence of 15.3% among men and 10.6% among women. Two systematic reviews and meta-analyses showed a prevalence of about 8.5% [10,11]. The first meta-analysis reported that being a woman was a significant risk factor for T2D [10], whereas the latest meta-analysis found a higher prevalence of T2D in men [11]. The meta-analyses, however, did not present the prevalence among men and women separately. The quality of the studies included in the meta-analyses was generally low.

There are sex differences in the occurrence of T2D across the life span. Women display significantly higher rates of T2D in youth, whereas men have a significantly higher prevalence of T2D in midlife [12]. Early-onset T2D among children and adolescents is an emerging phenomenon worldwide with a higher burden in Asia than in Europe and America combined [13,14,15,16,17]. Given the early onset of diabetes, many women are affected during fertile age. This may contribute to the accelerated rise in prevalence of T2D in young Asians, as the offspring of mothers with diabetes have increased risk of diabetes later in life [18].

Subjects with T2D have a twofold increased risk for cardiovascular disease (CVD), and CVD is the principal cause of death [19,20]. Notably, women who do not have diabetes are at a lower risk of developing CVD than men at the same age [21]. This pattern is reversed in the setting of T2D. Several large meta-analyses have shown that women with T2D exhibit a higher relative risk of incident coronary heart disease, fatal coronary heart disease, and stroke compared with their male counterparts [22,23].

Fasting plasma glucose (FPG) and oral glucose tolerance test (OGTT) have been used as diagnostic tools in almost all previous studies on prevalence of T2D in Nepal [10,11]. Glycated hemoglobin (HbA1c) has been demonstrated to be a highly specific and convenient alternative to FPG for diabetes screening [24] and has been recommended by WHO for screening and diagnosis of diabetes [25], with a cut-off of ≥6.5% (48 mmol/mol) as criterion for diabetes. There has been some disagreement about the use of HbA1c, due to concerns around the potential impact of ethnicity, and some countries in the Asia-Pacific region have chosen other cut-offs [26,27,28]. Lu et al. evaluated HbA1c for screening and diagnosis of T2D with OGTT as reference. They showed that HbA1c ≥  7.0% predicted the presence of T2D with 97.5% confidence when using OGTT as reference [29].

In this study we aimed to assess the prevalence of diabetes and associated risk factors among women aged ≥15 years in a rural district of Nepal using HbA1c with a cut-off ≥ 6.5% as a diagnostic tool. Additionally, HbA1c ≥ 7% was used as cut-off to increase the specificity. Moreover, we wanted to assess the awareness of diabetes in the study population.

## 2. Materials and Methods

### 2.1. Study Design, Study Site and Study Population

A cross-sectional survey was conducted during February 2012 to May 2013 among non-pregnant, married women in the Kavre district located in the central hilly region of Nepal. Reproductive health was the primary outcome and prevalence of overweight/obesity and diabetes the secondary outcome. The study site and population have been described previously in the studies focusing on reproductive health [30,31]. Briefly, all nonpregnant women ≥15 years of age who were or had been married were invited to a one-day screening. Written informed consent was obtained from those who agreed to participate in the study. Altogether, 1498 women consented to participate in the original study, but finally 1162 women were included in the analyses [30,31]. In the present study, women were included based on measurement of HbA1c. The participants were in general reluctant to give blood, and about 100 women declined.

### 2.2. Data Collection

Enrollment of women in the study was performed during 45 community visits. The government has recruited female community health volunteers (FCHVs) from the local community in all villages to carry out the National Maternal and Child-Health program at grassroots level through home visits. Prior to data and blood sample collection, 45 FCHVs, health workers and local leaders in the study area were informed about the study purpose and provided with community-based education on diabetes and risk factors. A questionnaire was administered and filled in (Appendix A) by trained female interviewers (auxiliary nurse midwifes and a public health officer). Data on socio-demographic characteristics were collected, including age, ethnicity, religion, occupation, education level, and household income. Moreover, information on the following was obtained: smoking habits (previous, current and never smoking), intake of instant noodles, milk and vegetarian diet; and family history of diabetes, hypertension, and CVD. In the analyses, intake of instant noodles and milk was categorized as intake ≥2 and <2 times a week. Waist circumference (WC), height and weight were measured, and body mass index (BMI) was calculated. Non-fasting blood samples were collected by a trained laboratory technician on EDTA tubes and were transported in a box containing ice to the Department of Biochemistry, Dhulikhel Hospital. Analyses of HbA1c were performed consecutively within 24 h by NycoCard Hemoglobin A1c test (Axis-shield, Norway). The assay was certified by the National Glycohemoglobin Standardization Program (NGSP).

### 2.3. Measurement of Anthropometric Variables

Height was measured by a measuring tape attached on the wall. Participants were requested to be barefoot, take off any hair ties and stood on a flat surface and looked straight forward without tilting their head. The height was measured in centimeters (cm). Weight was recorded in kilograms (kg) using a portable digital weighing scale, placed on a hard, flat surface. Individuals were asked to remove their footwear, put on light clothes, and stand on the scale to record their weight. BMI was calculated as weight in kg per square of height in meter. WC was measured in a separate room by female health workers, in standing position, at the end of a natural expiration, holding the arms relaxed at the sides and at midpoint between the lower margin of the last palpable rib in the mid axillary line and the top of the iliac crest (hip bone). The measurements were recorded in cm.

### 2.4. Definition of Variables

Diabetes was diagnosed if one of the following criteria was fulfilled: a positive history of diabetes, the use of antidiabetic medication, or HbA1c of ≥6.5% (48 mmol/mol), American Diabetes Association. Prediabetes was defined as HbA1c 5.7–6.4% (39–47 mmol/mol) [25,32]. In addition, diabetes prevalence was estimated according to the HbA1c cut-off of ≥7% as proposed by Lu et al. [29]. BMI was categorized according to the criteria for the Asian population recommended by WHO: underweight < 18.5 kg/m^2^, normal weight 18.5–23.0 kg/m^2^, overweight 23.0–27.5 kg/m^2^, and obese ≥ 27.5 kg/m^2^ [33]. Abdominal or central obesity was defined as WC ≥ 80 cm according to the criteria for Asian women [34].

### 2.5. Statistical Analysis

Continuous variables are presented as mean with standard deviation (SD) and categorical variables as counts and percentage. Group differences were analyzed using one-way ANOVA test with Dunnett’s post hoc test for comparison with continuous variables and Pearson’s Chi square test for categorical variables. Binary logistic regression analysis was performed to examine risk factors for the dependent variable diabetes. The results are presented as crude odds ratios (COR) and age-adjusted odds ratios (AOR), with 95% confidence intervals (CIs). All analyses were performed using the IBM SPSS statistics (version 28.0). Statistical level of significance (alpha) was set at <0.05.

### 2.6. Ethics Statement

The study was approved by the Nepal Health Research Council (Approval no. 124/2012), the Dhulikhel Hospital/Kathmandu University School of Medical Sciences Institutional Review Committee (Approval no. 38/2011) and the Regional Committee for Medical and Health Research Ethics, Central Norway (Ref. no 2011/2540, date of approval: 10 February 2012).

## 3. Results

A total of 757 women, mean ± SD 43.0 ± 14.0 years (range 17–86 years), had HbA1c successfully measured and were included in the current study. The fact that HbA1c was analyzed in only 50% of consenting women was mainly attributed to logistic challenges and lack of experience with large-scale studies in a remote district. As a consequence, collection of aliquots of whole blood necessary for HbA1c analysis was omitted in a substantial number. Additionally, 100 of the study subjects declined to give blood. Sociodemographic characteristics, lifestyle factors and anthropometrics were similar in women in whom HbA1c measurements were performed and those without HbA1c measurements (Appendix A).

### 3.1. Characteristics

Table 1 shows sociodemographic and lifestyle factors and corresponding mean HbA1c levels. Among the married women, 6.0% (45/747) were widowed and 0.1% divorced. The women belonged to three ethnic groups, Dalit, Adhivasi/Janajati and Brahmin/Chhetri. About 80% of the participants were from the Adhivasi/Janajati ethnic group, that may be referred to as “middle class”. Altogether, 10.8% of the women belonged to the Brahmin/Chhetri and 10.5% to the Dalit ethnicity, representing the advantaged and disadvantaged caste, respectively. Most of the women were uneducated. Agriculture was the primary source of income, however, only 198 women reported the source of income. The mean number of children per woman was 3.5 ± 1.9. About 1.5% (11/757) of the participants had a family history of diabetes and 3.2% (24/757) had a family history of CVDs and/or hypertension. Only 12% had heard about diabetes via TV, health workers, magazines, or radio. The majority of women (92.3%) were non-vegetarian, milk intake was reported by about 59.7% and instant noodles by 60.6%. Current smoking was reported by 26.2% (198/757), among them 48.0% smoked less than five cigarettes per day. Forty-six (6.0%) women had smoked previously, and 67.8% had never smoked. Mean BMI and mean WC were 22.8 ± 4.2 kg/m^2^ and 77.3 ± 9.1 cm, respectively (Table 1).

Mean HbA1c level increased with age, women >55 years displaying significantly higher level than the age group 17–34 years (6.1 ± 1.0% and 5.7 ± 0.8%, respectively, *p* < 0.001). Mean HbA1c level was significantly higher in subjects with consumption of instant noodles ≥2 times a week compared to <2 times (5.9 ± 0.9% and 5.7 ± 0.8%, respectively, *p* = 0.002). Women reporting milk intake ≥2 times weekly had higher HbA1c than those reporting intake <2 times, although borderline significant (5.9 ± 0.8% and 5.7 ± 0.9%, respectively, *p* = 0.052). HbA1c was also significantly higher among women with WC > 80 cm compared to those with WC < 80 cm (6.0 ± 1.1 and 5.7 ± 0.8%, respectively, *p* = 0.004).

### 3.2. Prevalence of Diabetes

Mean HbA1c ± SD in the total population was 5.8 ± 0.8%. As shown in Figure 1, a total of 13.5% (102/757) of the women had diabetes (HbA1c ≥ 6.5%) with a mean HbA1c of 7.3 ± 1.1%. When using a cut-off of HbA1c 7% (28, 34), the prevalence was 5.8%. Sixteen women had HbA1c level above 8%. Prediabetes was observed in 38.5% (292/757) (Figure 1). Only two women reported to have been diagnosed with diabetes previously. One of them had a high HbA1c level (10.6%) and received treatment with an oral drug, however, it was not reported which medication. The other was treated with lifestyle modification and had a HbA1c of 6.1%.

The characteristics of the study population according to diabetes status are shown in Table 2. The prevalence increased with age and was significantly higher in women aged >55 years compared to those 17–34 years, 11.0 and 22.2%, respectively. The overall prevalence of diabetes was higher in Brahmin/Chhetri ethnic group, although not significantly. Women reporting milk intake >2 times weekly had a higher prevalence of prediabetes and diabetes than those with intake <2 times weekly. Likewise, women with consumption of instant noodles >2 times weekly exhibited a higher prevalence of prediabetes and diabetes compared to intake <2 times.

### 3.3. Factors Associated with Diabetes

A bivariate logistic regression analysis was performed to identify risk factors associated with diabetes (Table 3). The odds of having diabetes increased with age, women ≥55 years displaying an OR of 2.3 (95% CI: 1.30–4.17) compared to the age group of 17–34 years. Instant noodle intake ≥2 times a week was associated with increased risk of diabetes compared to intake <2 times a week (OR 2.1, 95% CI: 1.37–3.21). Women with milk intake less than two times a week displayed a lower risk for diabetes than those reporting milk intake more than two times weekly (OR 0.5, 95% CI: 0.32–0.76). The odds of diabetes was 1.4 times higher among participants with waist circumference ≥ 80 cm in comparison to individuals with waist circumference < 80 cm, although not significant (*p* = 0.105). No association was seen between diabetes and BMI.

## 4. Discussion

To our knowledge, this is the first study addressing the prevalence of diabetes and associated risk factors among women specifically from a rural district in Nepal. In this study including 757 women aged 17 to 86 years, we observed a high prevalence of both diabetes and prediabetes, 13.5% and 38.5%, respectively. When using a cut-off of 7.0%, the prevalence of diabetes was 5.8%. Only two of the participants had been diagnosed with diabetes previously, and one of them was treated with an oral drug. In contrast to most previous studies in Nepal, we used HbA1c to diagnose diabetes. The prevalence increased with age but was high also in the youngest age groups. Consumption of instant noodles, milk intake and vegetarian diet were associated with increased risk for diabetes. WC was higher among women with diabetes compared to those without diabetes, albeit not significant. The awareness on diabetes was low, as only 12% had heard about diabetes.

We observed a diabetes prevalence of 13.5% among women in a rural setting based on HbA1c cut-off ≥ 6.5%. This prevalence was substantially higher than anticipated. Ideally, we should have measured FPG to confirm the diagnosis. As fasting blood samples were not available, we used a cut-off for HbA1c of 7%, as proposed by Lu et al. and Lim et al. to increase the specificity [29,35]. With this approach the prevalence declined to 5.8%. According to Lim et al., 95% of these subjects had diabetes based on FPG, whereas the remainder had IGT [35]. Thus, we may claim with a high degree of certainty that the majority of the 5.8% had diabetes. To verify whether those with HbA1c levels between 6.5–7% have diabetes, additional testing with FPG should be done.

Notably, FPG has been found to underestimate the rate of undiagnosed diabetes compared with HbA1c [36], as illustrated by two population-based studies in Vietnam and Korea, reporting a significant discordance between FPG and HbA1c measurements [35,36]. This has been shown in several Asian populations, but not in US adults [37,38,39,40].

We did not measure C-peptide or auto-antibodies and can therefore not differentiate between T1D and T2D. However, given the preponderance of T2D in Asia, it is reasonable that the majority of women in our study exhibited T2D. The fact that only two were diagnosed with diabetes previously is also in support of this, as subjects with T2D may be asymptomatic for a long period in contrast to individuals with T1D. This complies with studies showing that the undiagnosed proportion can be as high as 80–90% in some of the poorest settings [41,42].

Most previous studies on diabetes prevalence in Nepal have applied FPG and/or OGTT as diagnostic criteria [9,10,11]. We chose to use HbA1c, as blood sampling can be carried out in a non-fasting state and at any time of the day [43]. This was important for the feasibility of the study, as many of the participants had to walk for 1–2 h to reach the study site. In two recent studies conducted at high altitudes in the Mustang district of Nepal, HbA1c was also used to screen for diabetes. Aryal et al. observed an overall prevalence of 6.9% among 521 subjects living >2800 m [8]. Another survey by Koirala et al. including 188 participants ≥18 years of age residing at 3570 m showed a diabetes prevalence of 4.6% [44]. In the meta-analysis by Gyawali et al., the overall prevalence was 8.5%, whereas the rates in rural districts ranged from 0.03–2.5% [10]. Aryal et al. reported somewhat higher prevalence in rural settings, 2.9 and 3.3%, correspondingly [8]. That was also the case in the current study, diabetes being more prevalent than reported in rural settings previously, independent of the cut-off used for HbA1c. It should be recalled that our study site had a location that enabled visiting neighboring cities by bus in the course of a day. This may have affected the eating habits and facilitated a more semi-urban lifestyle. Our findings correspond with data from a study in semi-urban Nepal showing an overall prevalence of 11.7% and a prevalence of 10% in women [9]. A high diabetes prevalence of 7.9% was also demonstrated in a rural setting in Bangladesh applying HbA1c as a diagnostic tool [45].

We observed a surprisingly high rate of prediabetes by 38.5% in our study population. This is in line with Aryal et al. showing a prevalence of 39.3 and 36.8% in two rural settings at high altitudes [8]. Notably, they reported that prediabetes was more prevalent among those living in rural settings at 2890 m and 3270 m compared to urban settings [8]. Similarly, Koirala et al. found that 31.3% had prediabetes [44]. The prediabetes prevalence in the current study and the two studies at high altitudes was about three times higher than reported by Gyawali et al. in semi-urban Nepal [9].

In the study by Aryal et al., a total of 56.4% of participants living at 3270 m reported hazardous drinking [8]. There is strong evidence for a relationship between heavy drinking and the risk of diabetes [46,47]. Accordingly, the high prevalence of prediabetes among rural dwellers was interpreted as partly attributed to heavy drinking. In the present study, the majority belonged to the Adhivasi/Janajati ethnic group, that is also known to have a high alcohol intake [48,49]. The disadvantaged Dalit group displays an even more hazardous drinking pattern [50]. This could probably explain the very high rate of prediabetes of 60.8% among the 79 participants from the Dalit group. Unfortunately, alcohol-related information was not collected in our study population.

In concordance with previous studies the prevalence of diabetes and prediabetes increased by age. Indeed, the rate was high in all age groups, and more than a third of women in the youngest age group displayed prediabetes. This illustrates that women are prone to get T2D early and at a child-bearing age. This is of concern as exposure to in utero hyperglycemia seems to enhance the future risk of obesity and T2D in offspring [18]. This may be attributed to epigenetic changes induced by the intrauterine environment, and these changes could also be transferred to future generation, thus becoming intergenerational [51,52]. According to an expert panel of ADA, 70% of those with prediabetes will develop diabetes in the future [53]. Moreover, a recent meta-analysis including 29 studies and more than 10 million participants reported that prediabetes was associated with increased risk of all-cause mortality and CVD in the general population [54].

The increasing burden of diabetes is related to changes in dietary pattern and sedentary behavior leading to weight gain and overweight [7,9,55]. Asians have a higher percentage of body fat at any given BMI than Caucasians and are more disposed to T2D [56]. A lower cut-off for BMI and WC is therefore applied to define overweight/obesity and central obesity in Asians [14]. We observed a higher WC among women with diabetes, although not significant, whereas BMI did not differ between those with and without diabetes. This concords with studies showing a strong association between measures of abdominal obesity and development of T2D [57]. Notably, WC is a stronger predictor for diabetes than BMI in most populations [58]. Moreover, Asians are more prone to visceral fat accumulation compared with western populations, despite being generally less obese [59].

Previous studies in Nepal have shown differences in prevalence of overweight/obesity and diabetes between the ethnic groups. The highest prevalence has been reported in the Adhivasi/Janajati ethnic group [9,10,60,61]. Likewise, we observed that a higher proportion of women in this group had diabetes compared to the two other ethnicities, however, this was not significant.

People living in more remote districts have a lifestyle that make them less prone to diabetes. This is illustrated by three of the studies included in the meta-analysis by Gyawali et al., showing substantially higher prevalence rates in urban than in rural districts, 19.4%, 14.6%, and 1.4% versus 1.3%, 2.5%, and 0.03%, respectively [10]. Like other parts of Nepal, our study site had undergone a nutritional transition, as exemplified by a high intake of instant noodles. We observed that intake of instant noodles >2 days a week was associated with increased risk for diabetes. This finding was supported by a study including 10,711 South Koreans, 19–64 years of age, showing that women who consumed instant noodles ≥2 a week, were 68% more likely to develop glucose intolerance [62]. Vegetarian diet was also associated with increased risk for diabetes in the present study. This is in discordance with most previous studies. A vegetarian diet comprising whole plant foods has been shown to be most beneficial for diabetes prevention and management [62,63]. However, the South Asian diet is characterized by high intake of carbohydrates, trans fats, and saturated fats, which appear to promote the risk of T2D [64]. Rice is the main component of the vegetarian diet in Nepal. White rice constitutes up to 60% of the glycemic load among the Chinese and was found to be associated with an increased risk of diabetes in a meta-analysis [65]. Likewise, a study from India revealed a positive association between intake of white rice and risk of T2D [65,66]. Notably, as elaborated on above, consumption of instant noodles, another component of the vegetarian diet in Nepal, is also associated with increased risk for diabetes.

In contrast to most previous studies, we found that milk intake was associated with increased risk of diabetes. In a review, addressing intake of dairy products and T2D, it was concluded that intake of 3 servings of dairy products a day may exert a preventive effect on diabetes [67]. Intake of low-fat cheese and yogurt intake was especially associated with a lower T2D risk. Low-fat dairy products have been recommended in several dietary guidelines. In a comprehensive review by Drouin-Chartier et al. assessing several RCTs, no negative effect of dairy consumption was observed on cardiometabolic risk regardless of fat content [68]. In a Chinese cohort study including 45,411 men and women who were followed for 12 years, total intake of dairy products was associated with a significantly lower risk of T2D, whereas no association was observed between high-fat dairy and T2D [69,70,71]. The milk consumed in rural Nepal is not reduced in fat content, hence, based on previous studies, we would expect a neutral effect on risk for T2D in our study population.

The awareness on diabetes was very low in our study population. This was also reflected in the small number of women who had been diagnosed with diabetes and received treatment. In contrast, in the study by Gyawali et al. in semi-urban Nepal, 65% were aware of their disease, 94% of those who were aware received treatment, and 21% of the treated subjects had their diabetes under control [9]. Given that Nepalese women are the caretakers of the family, they are the key persons with respect to implementation of lifestyle changes. A meta-analysis that assessed the efficacy of lifestyle education for prevention of T2D in subjects at high risk, provided evidence that this approach reduced the incidence of T2D [72]. Thus, lifestyle education could be a useful tool for preventing T2D. There is a need for increased awareness at all levels and development of cost-effective measures to identify individuals at risk for diabetes. Appropriate management of those at risk is mandatory to decelerate the diabetes epidemic and reduce the burden of CVD.

The study has limitations. Logistic challenges and the fact that many women refused to give blood, resulted in a lower sample size and reduced the power of the study. Still, the study population is larger than in most previous studies in Nepal. The characteristics were similar in women who had HbA1c measured and those who had not, hence reducing the possibility that selection bias may have substantially distorted the estimates. The participation rate in the original study was 62% [30]. We do not have data on the women who did not participate and cannot exclude that these women differed from those who took part in the study. Thus, selection bias at that level cannot be ruled out. The findings may not be generalizable to women in all rural districts of Nepal and not to men. The information collected from the questionnaire relies on self-report and may be influenced by recall bias and social desirability bias. HbA1c should ideally have been measured twice. Moreover, it would have been an advantage if blood samples could have been collected in the fasting state, thus enabling analysis of FPG for comparison with HbA1c in the diagnosis of diabetes.

## 5. Conclusions

This study using HbA1c as a diagnostic tool disclosed a high prevalence of diabetes and prediabetes among women in a rural setting of Nepal. The awareness on diabetes was also very low. Aging and consumption of fast food like instant noodles were identified as some of the risk factors. The high prevalence among women in fertile age is of concern as the offspring of mothers with diabetes have increased risk for diabetes in the future. To combat the diabetes epidemic, there is an urgent need for increased awareness at all levels and development of cost-effective measures to identify individuals at risk for diabetes.

## Figures and Tables

**Figure 1 ijerph-19-07011-f001:**
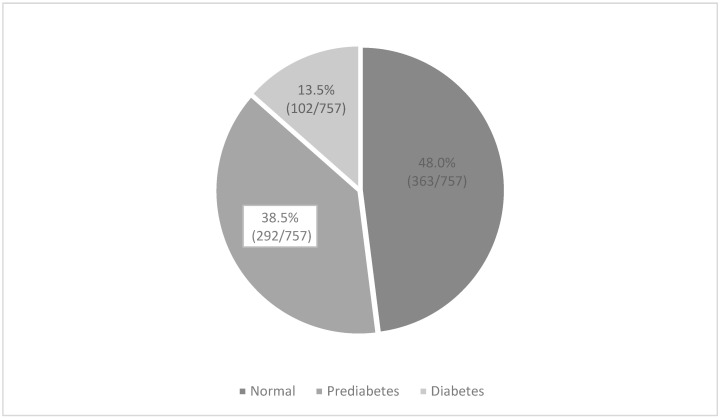
Prevalence of prediabetes and diabetes in the study population.

**Table 1 ijerph-19-07011-t001:** Sociodemographic and lifestyle factors by glycated hemoglobin (HbA1c) level of the study population.

Characteristics (*n* = 757)	*n* (%)/Mean ± SD	HbA1c Mean ± SD	*p* Value
Age (years)	43.0 ± 14.0		
HbA1c (%)	5.8 ± 0.8		
Age groups (years)			<0.001 #
17–34	210 (27.7)	5.7 ± 0.8	*
35–44	230 (30.4)	5.7 ± 0.8	0.860
45–54	173 (22.9)	5.7 ± 0.9	0.825
≥55	144 (19.0)	6.1 ± 1.0	<0.001
Ethnicity			0.639 #
Dalit	79 (10.5)	5.9 ± 0.7	
Adhivasi/Janajati	596 (78.7)	5.8 ± 0.9	
Brahmin/Chhetri	82 (10.8)	5.7 ± 1.1	
Educational status			0.798 #
Uneducated	663 (87.6)	5.7 ± 1.0	
Educated	94 (12.4)	5.8 ± 0.8	
Monthly household income ^a^ (NPR)			0.673 #
≤24,000	223 (30.5)	5.8 ± 0.9	
>24,000	507 (69.5)	5.8 ± 0.8	
Number of children	3.5 ± 1.9		
Parity			0.334 #
Null	21 (2.8)	5.6 ± 0.5	
1–3	377 (50.0)	5.8 ± 0.9	
>3	359 (47.2)	5.8 ± 0.9	
Dietary factors			
Vegetarian diet ^b^			0.858 #
Yes	35 (4.6)	5.8 ± 0.9	
No	721 (95.4)	5.9 ± 0.9	
Instant noodle intake			0.002 #
<2 times a week	506 (66.8)	5.7 ± 0.8	
≥2 times a week	251 (33.2)	5.9 ± 0.9	
Milk intake			0.052 #
≥2 times a week	289 (38.2)	5.9 ± 0.8	
<2 times a week	468 (61.8)	5.7 ± 0.9	
Smoking status			0.131 #
Current	198 (26.2)	5.9 ± 0.8	
Former	46 (6.0)	5.9 ± 1.0	
Never	513 (67.8)	5.7 ± 0.9	
Anthropometric measurements ^c^			
Height (cm)	149.3 ± 6.9		
Weight (kg)	50.7 ± 9.7		
BMI ^c^ (kg/m^2^)	22.8 ± 4.2		
BMI ^c^ (kg/m^2^), Asian cut-offs			0.224 #
Normal (18.5–22.9)	359 (50.0)	5.7 ± 0.8	
Underweight (<18.5)	66 (9.2)	5.8 ± 0.8	
Overweight/obese (>23.0)	292 (40.8)	5.8 ± 1.0	
WC ^d^ (cm), Asian cut-offs	77.8 ± 9.1		

Abbreviations: SD, standard deviation; NPR, Nepalese rupee; BMI, body mass index; WC, waist circumference; # overall *p* value by One way ANOVA; * reference group for Dunnett’s post hoc test; ^a^ missing *n* = 27; ^b^ missing *n* = 1; ^c^ missing *n* = 40; ^d^ missing *n* = 36.

**Table 2 ijerph-19-07011-t002:** Characteristics of the study population according to diabetes status.

Characteristics (*n* = 757)	Diabetes Status			
	Normal	Prediabetes	Diabetes	
	*n* = 363 (48)	*n* = 292 (38.5)	*n* = 102 (13.5)	*p* Value
HbA1c (%), mean ± SD	5.2 ± 0.4	6.0 ± 0.2	7.3 ± 1.1	
Age (years), mean ± SD	41.0 ± 12.6	44.0 ± 14.7	47.2 ± 15.2	<0.001
Age groups (years), *n* (%)				<0.001
17–34 (*n* = 210)	113 (53.8)	74 (35.2)	23 (11.0)	
35–44 (*n* = 230)	116 (50.4)	88 (38.3)	26 (11.3)	
45–54 (*n* = 173)	87 (50.3)	65 (37.6)	21 (12.1)	
≥55 (*n* = 144)	47 (32.6)	65 (45.1)	32 (22.3)	
Ethnicity, *n* (%)				<0.001
Dalit (*n* = 79)	22 (27.8)	48 (60.8)	9 (11.4)	
Adhivasi/Janajati (*n* = 596)	289 (48.5)	227 (38.1)	80 (13.4)	
Brahmin/Chhetri (*n* = 82)	52 (63.4)	17 (20.7)	13 (15.9)	
Educational status, *n* (%)				0.651
Uneducated (*n* = 663)	315 (47.5)	256 (38.6)	92 (13.9)	
Educated (*n* = 94)	48 (51.0)	36 (38.3)	10 (10.7)	
Number of children, mean ± SD	3.4 ± 1.8	3.6 ± 1.9	3.8 ± 1.8	0.068
Number of children, *n* (%)				0.283
Null (*n* = 21)	13 (62.0)	7 (33.2)	1 (4.8)	
1–3 (*n* = 377)	184 (48.8)	149 (39.5)	44 (11.7)	
<3 (*n* = 359)	166 (46.0)	136 (38.0)	57 (16.0)	
Dietary status				
Vegetarian diet ^a^, *n* (%)				0.233
Yes (*n* = 35)	16 (45.7)	11 (31.4)	8 (22.9)	
No (*n* = 721)	346 (48.0)	281 (39.0)	94 (13.0)	
Milk intake, *n* (%)				0.002
≥2 times a week (*n* = 290)	121 (41.5)	116 (40.1)	53 (18.3)	
<2 times a week (*n* = 467)	242 (52.0)	176 (37.5)	49 (10.5)	
Instant noodle intake, *n* (%)				<0.001
≥2 times a week (*n* = 251)	98 (39.0)	104 (41.5)	49 (19.5)	
<2 times a week (*n* = 506)	265 (52.5)	188 (37.0)	53 (10.5)	
Smoking status, *n* (%)				0.008
Current (*n* = 198)	76 (38.4)	94 (47.5)	28 (14.1)	
Former (*n* = 46)	18 (39.1)	21 (45.7)	7 (15.2)	
Never (*n* = 513)	269 (52.4)	178 (34.7)	67 (13.0)	
BMI ^b^ (kg/m^2^), mean ± SD	23.0 ± 4.4	22.3 ± 3.8	23.5 ± 5.5	0.024
WC ^c^ (cm), mean ± SD	77.6 ± 9.4	77.7 ± 8.7	79.2 ± 9.2	0.340

Abbreviations; HbA1c, glycated hemoglobin; SD, standard deviation; BMI, body mass index; WC, waist circumference; Data are in mean ± standard deviation (SD) or *n* (%). ^a^ missing *n* = 1; ^b^ missing *n* = 40; ^c^ missing *n* = 36.

**Table 3 ijerph-19-07011-t003:** Risk factors associated with diabetes in the study population.

Characteristics	Normal *n* (%)	Diabetes *n* (%)	COR ^a^ (95% CI)	*p* Value	AOR ^b^ (95% CI)	*p* Value
Age group, years						
17–34	188 (89.5)	23 (10.5)	1			
35–44	204 (88.7)	26 (11.3)	1.0 (0.57, 1.88)	0.907		
45–54	152 (87.9)	21 (12.1)	1.2 (0.60, 2.10)	0.717		
≥55	112 (77.8)	32 (22.2)	2.3 (1.30, 4.17)	0.005		
Ethnicity						
Adhivasi/Janajati	516 (86.6)	80 (13.4)	1			
Brahmin/Chhetri	69 (84.1)	13 (15.8)	1.2 (0.64, 2.30)	0.549	1.1 (0.60, 2.18)	0.682
Dalit	70 (88.6)	9 (11.4)	0.8 (0.40, 1.73)	0.617	0.8 (0.40, 1.76)	0.651
Educational Status						
Educated ^c^	84 (89.4)	10 (10.6)	1		1	
Uneducated	571 (86.1)	92 (13.9)	1.3 (0.67, 2.70)	0.391	0.9 (0.40, 1.84)	0.706
Household income per month (NPR)						
<24,000	193 (86.5)	30 (13.5)	1		1	
≥24,000	440 (86.7)	67 (13.3)	1.0 (0.65, 1.67)	0.938	1.0 (0.67, 1.72)	0.768
Vegetarian diet						
Yes	27 (77.0)	8 (23.0)	1		1	
No	628 (87.1)	93 (12.9)	0.5 (0.22, 1.15)	0.103	0.5 (0.22, 1.16)	0.108
Instant noodle intake						
No (<2 times per week)	453 (89.5)	53 (10.5)			1	
Yes (≥2 times per week)	202 (80.5)	49 (19.5)	2.1 (1.36, 3.16)	0.001	2.1 (1.37, 3.21)	0.001
Milk intake						
Yes (≥2 times per week)	236 (81.6)	53 (18.4)	1		1	
No (<2 times per week)	419 (89.5)	49 (10.5)	0.5 (0.34, 0.79)	0.002	0.5 (0.32, 0.76)	0.001
Current smoker						
No	485 (86.8)	74 (13.2)	1		1	
Yes	170 (85.9)	28 (14.1)	1.1 (0.68, 1.72)	0.749	1.0 (0.59, 1.54)	0.860
Waist circumference						
<80 cm	371 (87.5)	53 (12.5)	1		1	
≥80 cm	163 (82.7)	34 (17.3)	1.4 (0.91, 2.33)	0.113	1.4 (0.92, 2.37)	0.105

^a^ COR, crude odds ratio; ^b^ AOR, odds ratio adjusted for age; ^c^ educated defined as women who completed at least primary school (5 years school).

## Data Availability

Not applicable.

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
