# Peer review of "Diabetes Prevalence and Associated Risk Factors among Women in a Rural District of Nepal Using HbA1c as a Diagnostic Tool: A Population-Based Study"

_ijerph, 2022, doi:10.3390/ijerph19127011_

Round 1
Reviewer 1 Report
Dear Authors,
Thank you for your article. I think the most important lack of information is the control group here.
-Why did not choose any control group for your experiments?
-How did you choose the patients? (according to which criteria)
-What is the sufficient volume in line 97?
-There are too many tables in this manuscript. Please combine some of them to emphasize main purpose of the study.
-You analyzed many risk parameters but you didnt write the statistics detaily. How did you analyze and emphasize these parameters effects? Is there dependent or independent?
-Where are the statistical test results?
You should also expand the conclusion part about the significance of this study to readers. You need to highlight it.
Author Response
Dear Reviewer 1
Thank you for providing an opportunity to address your comments and suggestions. We have addressed the comments and revised the manuscript accordingly. We find that the comments have contributed to improvement of the manuscript.
We have addressed the comments with answers (in black and bold) font and changes in the revised manuscript.

Reviewer 2 Report
Keeping in view the remoteness of the region it is a good effort, but it needs further imbibement. Kindly consult the following literature in order to improve this research:
- Furukawa, T., Nishida, Y., Hara, M., Shimanoe, C., Koga, K., Iwasaka, C., Higaki, Y., Tanaka, K., Nakashima, R., Ikezaki, H. and Hishida, A., 2022. Effect of the interaction between physical activity and estimated macronutrient intake on HbA1c: population-based cross-sectional and longitudinal studies. BMJ Open Diabetes Research and Care, 10(1), p.e002479.
- Rohlfing, C.L., Little, R.R., Wiedmeyer, H.M., England, J.D., Madsen, R., Harris, M.I., Flegal, K.M., Eberhardt, M.S. and Goldstein, D.E., 2000. Use of GHb (HbA1c) in screening for undiagnosed diabetes in the US population. Diabetes care, 23(2), pp.187-191.
- Sample size is quite small. Any particular reason?
- Why the other glucose monitoring tests not done and compared, if you want to establish HbA1c as gold standard?
- Due to the dynamic nature of diabetes, what will be the significance of this almost decade old study? What will be its novelty?
- Any particular reason/s for not using controls?
Author Response
Dear Reviewer 2
We sincerely appreciate your valuable comments and suggestions. We find that the comments are relevant and contributed to improvement of the manuscript. We have addressed the comments and revised the manuscript accordingly.
We have addressed the comments with answers (in black and bold) font and changed in the revised manuscript.

Reviewer 3 Report
- Abstract : “A novelty of the study is the use of HbA1c as diagnostic criterion.” How can this conclusion be drawn?
There are many studies related to HbA1c, please refer to them. Such as:
“HbA1c and birthweight in women with pre-conception type 1 and type 2 diabetes: a population-based cohort study”
“HbA1c as a diagnostic tool for diabetes and pre-diabetes: the Bangladesh experience”
- The analysis of this paper is too simple and the significance analysis is not comprehensive.
Author Response
Dear Reviewer 3
We sincerely appreciate your valuable comments and suggestions. We find that the comments are relevant and contributed to improvement of the manuscript. We have addressed the comments and revised the manuscript accordingly.
We have addressed the comments with answers (in black and bold font) and changes in the revised manuscript.

Reviewer 4 Report
This is a well-written and interesting article evaluating diabetes prevalence and associated risk factors among women in Nepal; minor revisions are suggested before possible publication.
Line 105: it is said that “questionnaire was administered and filled in (supplementary files)”, but the questionnaire cannot be found in the supplementary files; please revise.
Moreover, a thorough English revision may be needed (line 122: in kilograms; line 124 clothes; line 231 cut-off…)
Author Response
Dear Reviewer 4
Thank you for your valuable comments and suggestion. The comments are very helpful to improve our manuscript and we have addressed the comments received from revision of the manuscript. We have addressed the comments with answers (in black and bold font) and have changes in the revised manuscript.

Round 2
Reviewer 1 Report
Dear Authors,
Thank you for your revision. It is better than before. I think it should be accepted as it is.
Best regards,
Reviewer 2 Report
The authors have largely addressed my concerns, I appreciate their efforts.
Reviewer 3 Report
Ok